# Low Energy Availability Risk Is Associated with Anxiety in Female Collegiate Athletes

**DOI:** 10.3390/sports12100269

**Published:** 2024-10-08

**Authors:** Jennifer L. Scheid, Sabrina Basile, Sarah L. West

**Affiliations:** 1Department of Physical Therapy, Daemen University, Amherst, NY 14226, USA; 2Department of Kinesiology and Biology, Trent University, Peterborough, ON K9L 0G2, Canada

**Keywords:** Female Athlete Triad, RED-S, reproductive dysfunction, eating disorder, low energy availability, menstrual cycle disturbance

## Abstract

This study investigated the association between the risk of low energy availability, disordered eating, and anxiety in collegiate female athletes. Female athletes (*n* = 115) completed questionnaires that assessed disordered eating (Disordered Eating Screen for Athletes, DESA-6; and the Eating Disorder Examination Questionnaire Short, EDE-QS), anxiety (Generalized Anxiety Disorder-7) and the risk of low energy availability (Low Energy Availability in Females Questionnaire; LEAF-Q). The athletes were 19.9 ± 0.1 years old and presented with no anxiety (14.8%), mild (33.0%), moderate (24.3%), and severe (27.8%) anxiety. The EDE-QS scores revealed that 22.6% of the participants had a high risk of an eating disorder, while the DESA-6 scores revealed that 31.3% of the participants scored positive for a risk of disordered eating. The LEAF-Q total scores revealed that 68.7% of the participants were at risk of low energy availability. Increased GAD-7 scores were associated (*p* < 0.001) with measures of disordered eating (EDE-QS and DESA-6) and the risk of low energy availability (LEAF-Q total score). Non-parametric partial correlations demonstrated that anxiety (increased GAD-7 scores) correlated with the risk of low energy availability (increased LEAF-Q total scores) while controlling for eating disorder scores (EDE-QS) (r (112) = 0.353, *p* < 0.001), or while controlling for the risk of disordered eating (DESA-6 scores) (r (112) = 0.349, *p* < 0.001). In female collegiate athletes, both disordered eating and the risk of low energy availability were positively associated with increased anxiety.

## 1. Introduction

In the early 1990’s, the Female Athlete Triad was conceptualized and defined as a decreased bone mineral density, menstrual dysfunction, and decreased energy availability [1,2,3,4]. In 2014, the term Relative Energy Deficiency in Sport (RED-S) was developed for low energy availability in athletes and was used to provide a model for the plethora of potential effects of low energy availability [5]. Research has demonstrated that the Female Athlete Triad and RED-S occur in athletes as a result of low energy availability (with or without disordered eating), leading to impairments in physiological function in any of the following areas: the cardiovascular system, immunity, menstruation, bone mineral density, metabolic rate and protein synthesis [4,5]. While mechanisms linking low energy availability to both menstrual dysfunction and a low bone mineral density have been well researched [6,7,8,9], some of the framework within RED-S is not as well understood. Mental health, defined as a person’s psychological well-being, is one area with little research related to low energy availability, despite the 2023 International Olympic Committee’s (IOC) consensus statement on Relative Energy Deficiency in Sport (REDs) highlighting a bidirectional relationship between mental health issues and low energy availability [10].

Collegiate-level athletes are a unique group that are at risk of reduced mental health [11] related to the balancing of advanced sport participation with academic performance, among other factors. For example, recent research on the prevalence of alcohol use and anxiety or depression in current elite-level athletes (ages 16–29) reported values of 19% and 34%, respectively [12]. A relationship exists between the mental and physical health of an individual [13], and an athlete experiencing symptoms of overtraining and burnout may display symptoms such as physical and emotional exhaustion [14]. Anxiety, which is an increased feeling of dread or uneasiness, has been found to be associated with Female Athlete Triad risk [15,16,17,18] and RED-S [19]. In one study, menstrual irregularity and anxiety were associated in a group (*n* = 104) of elite rowers (aged 16–23) who were on the national team, preparing for international competitions [18]. More recently, a study of National Collegiate Athletics Association (NCAA) Division I (DI) female cross country runners demonstrated that high anxiety may commonly present with other symptoms of RED-S [19]. However, there is limited research in this area and therefore more research on low energy availability and its association with the psychological well-being of college athletes is needed. Understanding the relationship between aspects of the Female Athlete Triad/RED-S and aspects of mental health, like anxiety, can help inform the prevention of these issues and the clinical management of athletes. The aim of the current study was to explore the association between the risk of low energy availability, disordered eating, and general anxiety in female college athletes. We hypothesized that the risk of low energy availability would be positively associated with increased anxiety scores and that increased disordered eating would also be positively associated with increased anxiety.

## 2. Materials and Methods

### 2.1. Participants

This study invited current female athletes participating in college-level athletics in Division I, II, or II of the National Collegiate Athletics Association (NCAA) to participate. Athletes had to be at least 18 years of age and a student currently participating in a collegiate sport on campus. The institutional review board approved the study, and informed consent was obtained from all the participants via the survey tool prior to study participation.

Online questionnaires were distributed via email to college athletes through athletic trainers, coaches, and athletic directors. The emails contained a link to an anonymous questionnaire. Athletes chose to participate in the study. To participate in the study, the athletes had to answer “yes” to their willingness to participate in the study, “yes” to being at least 18 years or older, and “yes” to being an NCAA athlete.

### 2.2. Data Collection

An online questionnaire was created using the Google Forms web application. This questionnaire consisted of general demographic questions, the Disordered Eating Screen for Athletes (DESA-6) [20], the Eating Disorder Examination Questionnaire-Short (EDE-QS) [21,22] the Low Energy Availability in Females Questionnaire (LEAF-Q) [23] and the General Anxiety Disorder-7 (GAD-7) [24].

### 2.3. Measurements

#### 2.3.1. Demographics

The athletes self-reported their gender, sport, age, body mass and height. The athletes’ body mass index (BMI) was calculated based on their self-report body mass and self-reported height.

#### 2.3.2. Disordered Eating

Disordered eating was measured using the Disordered Eating Screen for Athletes (DESA-6). DESA-6 is a six-item screening tool that was designed for athletes of both genders, with the purpose of determining the athletes’ risk of disordered eating [20]. The questionnaire asks general health questions, and questions pertaining to injuries in previous sports seasons, weight, and dieting to determine the risk of the athlete participating in disordered eating. The DESA-6 is scored out of 6, with a total score of 3 or greater indicating the presence of disordered eating. The DESA-6 has been demonstrated to be a reliable and valid instrument for measuring the risk of disordered eating in adolescent athletes [20] and has been used to measure the risk of disordered eating in university athletes [25]. The DESA-6 scale score has construct validity, demonstrated by its correlation with the reference scale (r = 0.8), as well as its test–retest reliability (correlation r = 0.8) [20].

#### 2.3.3. Risk of an Eating Disorder

While the main goal of this study was to focus on the relationship between disordered eating and/or low energy availability in college athletes, we also measured the risk of extreme disordered eating, i.e., the risk of an eating disorder. Eating disorder risk was measured using the Eating Disorder Examination Questionnaire-Short (EDE-QS) [21]. The EDE-QS consists of 12 items that capture various aspects of eating disorder psychopathology, including dietary restraint, eating concern, shape concern, and weight concern over the preceding 7 days. The total score on the EDE-QS ranges from 0 to 36, with a higher score indicating more disordered eating behaviors and a score of 15 or higher indicating the risk of an eating disorder. The EDE-QS had been shown to be a reliable and valid in measuring the risk of an eating disorder in adults [21,22]. The EDE-QS scale score has internal consistency (Cronbach α = 0.91), as well as test–retest reliability (intraclass correlation = 0.93) [22].

#### 2.3.4. Risk of Low Energy Availability

The Low Energy Availability in Females Questionnaire (LEAF-Q) was used to measure the risk of low energy availability. The LEAF-Q is a questionnaire developed to identify female athletes ‘at risk’ of the physiological symptoms associated with low energy availability, including injuries, gastrointestinal issues, and reproductive function [23]. The questionnaire is divided into three sections pertaining to different areas related to the symptoms of low energy availability: injuries, gastrointestinal function, and menstrual function/contraceptive use. The scores from each section are added together for a total LEAF-Q score. The total LEAF-Q scores range from 0 to 49, with a higher score (greater or equal to 8) indicating a greater clinical risk of low energy availability and, therefore, a greater risk of the Female Athlete Triad and RED-S [23]. The LEAF-Q scale score has internal consistency (Cronbach α = 0.86), as well as test–retest reliability (intraclass correlation = 0.79) [23].

#### 2.3.5. Symptoms of Generalized Anxiety Disorder

The Generalized Anxiety Disorder-7 (GAD-7) was used to measure anxiety. The GAD-7 is a seven-item self-report questionnaire designed to determine mental health symptoms related to Generalized Anxiety Disorder (GAD) over a two-week period [24]. Questions focus on the participants’ feelings of nervousness and worry, their ability to relax, and various other symptoms to determine the overall anxiety severity in the participant. The total score on the GAD-7 ranges from 0 to 21, with higher scores indicating greater anxiety severity. The cut-off scores for the GAD-7 are as follows: (1) a score of 0 to 4 indicates minimal or no anxiety symptoms, (2) a score of 5 to 9 indicates mild anxiety symptoms, (3) a score of 10 to 14 suggests moderate anxiety symptoms, indicating a moderate level of distress and the potential presence of generalized anxiety disorder, and (4) a score of 15 or higher suggests severe anxiety symptoms, indicating a high level of distress and a higher likelihood of meeting the criteria for GAD. The GAD-7 has been shown to be reliable and valid in measuring anxiety in adults [24], and has been used to measure the risk of disordered eating in college athletes [19]. The GAD-7 scale score has internal consistency (Cronbach α = 0.92), as well as test–retest reliability (intraclass correlation = 0.83) [24]. In addition to categorizing the athletes by minimal, mild, moderate, or higher anxiety (to report frequency of these categories), the demographic information and study outcomes were presented in two groups: minimal/moderate anxiety and moderate/high anxiety. These groups were created to simplify the presentation of these data.

### 2.4. Data Analysis

Descriptive statistics were used to examine the athletes’ demographic characteristics. The Kolmogorov–Smirnov test was used to determine if the data were normally distributed. Since GAD-7, EDE-QS, and DESA-6 were not normally distributed, non-parametric tests were used. The independent samples Mann–Whitney U test was used to compare the demographic characteristics, general anxiety, disordered eating, and risk of low energy availability in athletes with no/mild anxiety to athletes with moderate/severe anxiety (determined by the GAD-7). Spearman’s correlation was used to determine the correlation between anxiety (GAD-7) and disordered eating (EDE-QS and DESA-6) and the risk of the Female Athlete Triad (LEAF-Q). Non-parametric partial correlations were used to determine if the correlation between anxiety (GAD-7) and the risk of the Female Athlete Triad (LEAF-Q) was significant while controlling for disordered eating (EDE-QS or DESA-6). A significance level of *p* < 0.05 was used to identify all significant differences, and all data were analyzed using the SPSS (version 28.0; Chicago, IL, USA) statistical software package.

## 3. Results

### 3.1. Demographic Characteristics

A total of 115 female athletes consented to participate in the study. The mean age of the participants was 19.9 ± 0.1 years, and the mean body mass index (BMI) was 23.3 ± 0.2 kg/m^2^. Participants played the following sports: basketball (11.3%), bowling (3.5%), cross country/track and field (20.9%), lacrosse (34.5%), rugby (0.9%), soccer (23.5%), softball (0.9%), swimming (0.9%), tennis (0.9%) and volleyball (2.6%). Participants played at the NCAA Division I (14.8%), Division II (56.5%), and Division III (28.7%) level. There was no difference (*p* > 0.05) in age, body weight, or BMI when comparing participants in the no/mild anxiety group to the moderate/severe anxiety group (Table 1).

### 3.2. Anxiety, Eating Disorder Risk, Disordered Eating, and Low Energy Availability Risk

The participants presented with no anxiety (14.8%), mild anxiety (33.0%), moderate anxiety (24.3%), and severe anxiety (27.8%). The EDE-QS scores revealed that 22.6% of the participants had a high risk of an eating disorder (77.4% had normal scores). DESA-6 scores revealed that 31.3% of the participants scored positive for the risk of disordered eating (68.7% had normal scores). The LEAF-Q total scores revealed that 68.7% of the participants were at risk of the Female Athlete Triad (31.3% had no risk). Participants in the moderate/severe anxiety group had higher GAD-7 scores (15.3 ± 0.5 vs. 5.7 ± 0.3, *p* < 0.001), EDE-QS scores (11.9 ± 1.1 vs. 5.2 ± 0.8, *p* < 0.001), DESA-6 scores (2.5 ± 0.2 vs. 1.2 ± 0.2, *p* < 0.001), and LEAF-Q total scores (11.7 ± 0.6 vs. 8.2 ± 0.5, *p* < 0.001) compared to participants in the no/mild anxiety group, i.e., the athletes with moderate/severe anxiety also demonstrated a higher risk of disordered eating and eating disorders (Figure 1).

### 3.3. Correlations with Anxiety

Table 2 presents the correlation data. A significant positive correlation between the anxiety score (GAD-7) and EDE-QS score (*p* < 0.001) and the DESA-6 (*p* < 0.001) was found. Anxiety was also correlated with the LEAF-Q total score (*p* < 0.001), injury subscale (*p* = 0.004), and gastrointestinal subscale (*p* < 0.001). However, the LEAF-Q menstrual scores did not correlate with anxiety (*p* = 0.173).

Non-parametric partial correlations demonstrated that anxiety (GAD-7 scores) correlated with the risk of low energy availability (LEAF-Q total score) while controlling for the eating disorder score (EDE-QS) (r (112) = 0.353, *p* < 0.001), and that anxiety (GAD-7 scores) correlated with the risk of low energy availability (LEAF-Q total score) while controlling for the risk of disordered eating (DESA-6) (r (112) = 0.349, *p* < 0.001), demonstrating that the risk of low energy availability and disordered eating are independently associated with anxiety in college athletes.

## 4. Discussion

The current cross-sectional study examined the associations between the risk of low energy availability and anxiety, as measured by the GAD-7 in 115 female collegiate athletes. We found that over half of the participants presented with moderate–severe anxiety and that this increased disordered eating tendencies (as measured by the EDE-QS and the DESA-6) and the risk of low energy availability (as assessed by LEAF-Q menstrual score subscale), which was positively correlated with anxiety in female athletes. In addition, for the first time, we demonstrated that the risk of low energy availability and disordered eating were independently associated with anxiety in college athletes. While the relationship between anxiety and eating disorders has been documented since in 1950s [26], demonstrating that low energy availability is related to anxiety, independent of the risk of eating disorders, this is an important step in understanding the complex relationship between the Female Athlete Triad/RED-S and anxiety.

Our findings are similar to those of other published research in the literature [15,16,17,18,19]. For example, Wolfenden and colleagues [18] found significant positive correlations between measures of depression and anxiety and increased Triad scores, as assessed using the Female Athlete Triad Coalition Cumulative Risk Assessment Tool in 78 female high school athletes. In this cross-sectional study, anxiety was assessed using the Patient-Reported Outcomes Measurement Information System Emotional Distress Depression and Anxiety short-form questionnaires and was associated with factors of the Female Athlete Triad. In another study of 104 elite rowers (Japan national team) aged 16 to 23 years, factors related to mental health characteristics such as state and trait anxiety, and physical characteristics such as diet, body mass and composition, and energy availability, were used to determine their interaction with participant self-reported menstrual cycles [17]. The authors found that the athletes’ state of anxiety (measured by multiple logistic analysis using the State and Trait Anxiety Inventory) was predictive of menstrual cycle irregularities [17]. More recently, Olsen and colleagues [16] also found a correlation between the Female Athlete Triad (measured by the Female Athlete Triad Coalition Cumulative Risk Assessment Tool) and anxiety symptoms in 780 athletes (76% post-collegiate, average age ~29 years old). While these studies included female athletes at both the amateur and elite level of sport across a variety of ages, our study is novel in that we report the association between the risk of low energy availability and increased anxiety in female athletes at the collegiate level. Additionally, in the current study, we demonstrated that disordered eating tendencies and the risk of low energy availability are independently related to anxiety.

One possible mechanism that explains our observed correlation between anxiety and the Female Athlete Triad (measured by the risk of low energy availability) is the dysregulation of sex hormones. Low energy availability and therefore the risk of the Female Athlete Triad may be associated with low or insufficient sex hormone levels (i.e., estrogen and progesterone), and studies have shown that sex hormones can have an impact on mental health. For example, Gouveia and colleagues examined sex hormone changes in a rodent model and found that anxiety behaviors were associated with estrogen fluctuations [27]. In a systematic review, Dubol and colleagues analyzed the neuroimaging of 1304 women, and reported that the modulation of estrogen and other hormones led to changes in the cortico-limbic brain regions, which suggests that fluctuations in hormones affect brain structure and may impact mental health [28]. In a review, Kundakovic and Rocks discussed human and animal models that provide evidence that sex hormone fluctuations (and estrogen withdrawal) increase the risk of depressive symptoms and disorders [29]. We hypothesize that hormonal changes caused by a low energy availability could simultaneously cause anxiety symptoms, and/or vice versa.

Another potential mechanism that connects anxiety symptoms to the Female Athlete Triad is how the body responds to chronic stresses associated with sport performance, i.e., stress may mediate the relationship between anxiety and low energy availability. Athletes are well known to experience stress with high-intensity training. Clark and Mach discussed how stress is a natural adaptation for humans in order to maintain homeostasis [30]. However, it also can be maladaptive and induced by outside challenges. Psychological stress is important to consider for individuals with existing mental health conditions or those that lack appropriate stress management strategies. Ranabir and Reetu examined how psychological stress causes the release of hormones such as cortisol [31]. After a stressful stimulus, these stress-related hormones typically decrease as the body re-adjusts to baseline or homeostasis. However, in a state of continued psychological stress, the chronic stimulus may maintain the elevated hormone response. Stress responses include increases in the corticotropin-releasing hormone, which can decrease the luteinizing hormone and follicle stimulating release by inhibiting the gonadotropin-releasing hormone in the hypothalamus [32]. In 2010, Pauli and Berga described neuroendocrine alternations that cause athletic amenorrhea to be a synergistic combination of metabolic and psychogenic stress [33]. In this context, it might be possible that the signs of low energy availability (including menstrual cycle disturbances) observed in the Female Athlete Triad may exacerbate the psychological stress triggered by a chronic stress-related hormonal response. Further research examining the stress regulation activities of athletes to see if stress mediates (or moderates) the relationship between anxiety and the Athlete Triad is needed.

As previously noted, over a quarter (~28%) of our participants experienced severe anxiety. The reported prevalence of severe anxiety in this study is greater than the general US adult population (19.1%) but is less than reports in other elite athletes (34%) [12,34]. The results of our study indicate that female athletes experiencing symptoms of the Athlete Triad may benefit from interventions that target reductions in anxiety. Mental health professionals should be included in the continuum of care for athletes and, in addition to screening athletes for the symptoms of the Female Athlete Triad using the Female Athlete Triad Cumulative Risk Assessment [4] or low energy availability using the LEAF-Q [23,35], mental health screening should be employed regularly to identify athletes with symptoms of, and at risk of developing, mental health syndromes [36]. Proactive practices that identify risk factors and provide early intervention should be prioritized. For example, athletes who are identified as at risk of anxiety should be referred to a mental health professional for psychotherapy and other potential treatments (e.g., medications in more severe cases). Athletes who are identified as at risk for low energy availability or the Female Athlete Triad should be referred to a sports medicine physician and/or a sports nutritionist who has experience working with female athletes. It may be necessary to increase daily energy intake (modestly 300–350 kcal/day) to reverse the amenorrhea or oligomenorrhea that is often observed in female athletes with low energy availability [37]. Effective education, screening, and referral for athletes with at least one positive component of the Triad are an ongoing necessity in order to minimize future complications, including negative mental health and wellbeing outcomes.

It is important to note that our study had several limitations. The participants were only 115 female collegiate athletes who volunteered to participate in our survey. This limits the generalizability of our study to all female athletes or to the male athlete population. This study only represents one time point, i.e., the athletes were not monitored over time. In future studies, it would be important to replicate the study with a larger sample size, including males and different sporting levels and following the athletes for longer time periods to attempt to capture the development of physical and psychological pathologies over time. In our current study, we only examined one outcome of mental health: the severity of the symptoms of anxiety (which may be transient, as they were measured over the prior 2 weeks), and we did not address clinically diagnosed anxiety. Discriminating between the symptoms of anxiety and clinical anxiety will be important for future studies and will help inform future interventions. Anxiety is not the only mental health condition that athletes may have, and our study did not examine the relationship between other conditions such as depression or post-traumatic stress disorder (PTSD) [12,38]. Gouttebarge et al. found that 34% of current athletes from a population of 2895 athletes and 26% of 1579 former athletes had instances of anxiety/depression [12]. Lynch estimated that 1 in 8 elite athletes suffer from PTSD, and this disorder can be difficult to treat [38]. It is important for future studies to analyze how those conditions might also influence or be influenced by low energy availability. The current study is cross-sectional in nature; therefore, it is not possible to establish the causal direction between low energy availability and anxiety. Currently, not enough research exists to reliably ascertain the directionality of causation, but research on hormonal fluctuations suggests that the Female Athlete Triad may cause anxiety; alternatively, theories on psychogenic stress suggest that stress related to anxiety can worsen low energy availability and therefore impact the Female Athlete Triad. It is important for future research to determine the causative relationship between anxiety and the Female Athlete Triad symptomatology. Lastly, the current study relies on self-reported data. It can be expected that the athletes’ body weight and BMI in the demographic data are underreported [39]. Additionally, some of the questions in the survey were sensitive in nature, i.e., questions about anxiety, eating habits, and reproductive health, which may lead the proportion of athletes at risk of anxiety, eating disorders and/or low energy availability to be underreported in this study.

## 5. Conclusions

In the current study, over half of the collegiate female athletes displayed moderate to severe anxiety, and the risk of the Female Athlete Triad or risk of RED-S was associated with worse anxiety symptoms. More specifically, both disordered eating and the risk of low energy availability were associated with worse anxiety symptoms. These findings justify the ongoing need to screen athletes for risk factors for the Female Athlete Triad and state anxiety. In addition, future studies examining conditions such as the Female Athlete Triad need to account for non-standard risk factors such as mental health for a more complete picture of the condition. Further research is needed to differentiate the risk factors for athletes, including the level of athletic participation, the type of sport, and the presence of other mental health conditions. Research may include an analysis of the effects of improving or deteriorating mental health, the type and level of sport participation, and the influence of various mental health interventions. The findings of this study lay the groundwork for future studies that establish the long-term implications of these aspects of the Female Athlete Triad/RED-S for female athletes who have anxiety by providing an initial framework for understanding the correlational relationship of these factors.

## Figures and Tables

**Figure 1 sports-12-00269-f001:**
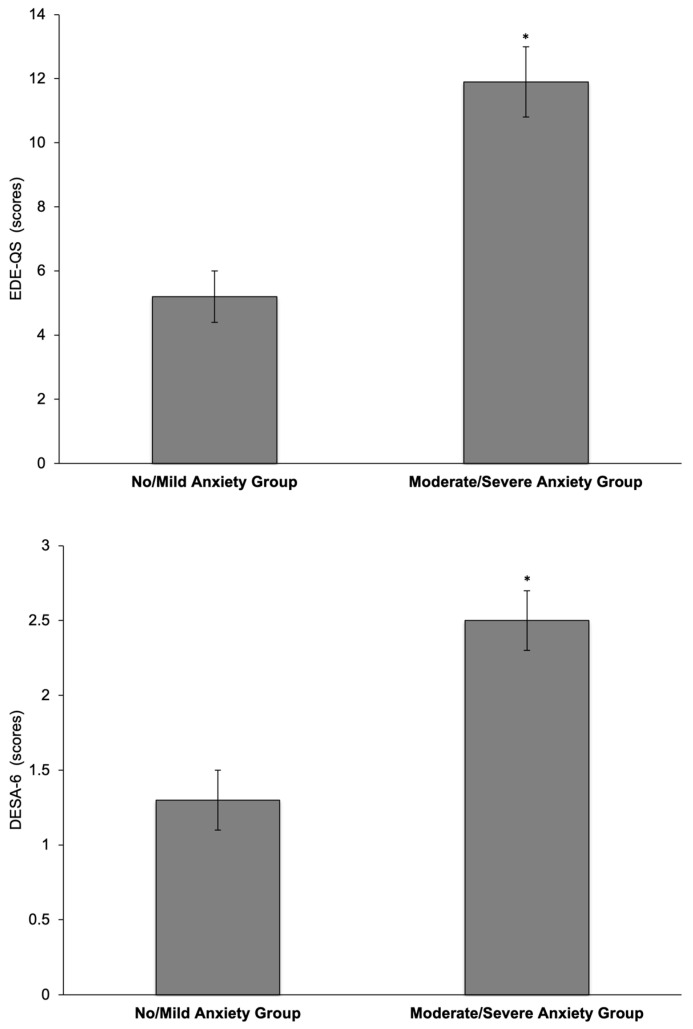
Scores indicating the risk of eating disorders (EDE-QS) and the risk of disordered eating (DESA-6) in female athletes with no/mild anxiety (*n* = 55) and moderate/severe anxiety (*n* = 60). * *p* = <0.001 when comparing groups. Data are reported as mean ± SEM. EDE-QS = Eating Disorder Examination Questionnaire-Short, DESA-6 = Disordered Eating Screen for Athletes.

**Table 1 sports-12-00269-t001:** Age, body weight, and BMI of participants with no/mild anxiety and moderate/severe anxiety.

	No/Mild Anxiety (*n* = 55)	Moderate/Severe Anxiety (*n* = 60)	*p*-Value
Age (years)	19.9 ± 0.2	19.9 ± 0.2	0.978
Body Weight (kg)	64.9 ± 1.4	65.8 ± 1.3	0.625
Body Mass Index (kg/m^2^)	22.9 ± 0.3	23.6 ± 0.4	0.140

* *p* < 0.05 considered significant when comparing groups. Data are reported as mean ± SEM.

**Table 2 sports-12-00269-t002:** Spearman correlations between disordered eating, low energy availability and generalized anxiety.

	General Anxiety Disorder-7	
	r_s_ Value	*p* Value
Disordered Eating		
EDE-QS Score	0.465	<0.001 *
DESA-6 Score	0.391	<0.001 *
Risk of the Female Athlete Triad		
LEAF-Q Total Score	0.430	<0.001 *
LEAF-Q Gastrointestinal Score	0.373	<0.001 *
LEAF-Q Menstrual Score	0.128	0.173
LEAF-Q Injury Score	0.264	0.004 *

* *p* < 0.05 considered significant, EDE-QS = Eating Disorder Examination Questionnaire-Short, DESA-6 = Disordered Eating Screen for Athletes, LEAF-Q = Low Energy Availability in Females Questionnaire. Note: All relationships were positive (+).

## Data Availability

The data presented in this study are available upon request from the corresponding author.

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
