# Peer review of "Low Energy Availability Risk Is Associated with Anxiety in Female Collegiate Athletes"

_sports, 2024, doi:10.3390/sports12100269_

Round 1
Reviewer 1 Report
Comments and Suggestions for Authors
Thank you for giving me the opportunity to review this intriguing work. The topic addressed, which explores the link between reduced energy availability and anxiety in female collegiate athletes, is of great relevance to both sports research and the mental health of athletes. The care with which the study was conducted is evident from the rigorous methodology and presentation of the results. The work is distinguished by the clarity of the exposition and the scientific relevance of the subject matter. In particular, the focus on the association between anxiety and disordered eating behaviour in female athletes adds an important piece to the understanding of the Female Athlete Triad and RED-S. The use of validated instruments such as the GAD-7, the LEAF-Q and the DESA-6 for data collection is also appreciable, reinforcing their scientific validity.
Suggested modifications:
Lines 3-7 (author information and correspondence): Add any contacts or information to facilitate direct access to the data, if possible. In addition, specify if there are any further contributions, such as a figure managing the data.
Line 8 (abstract): The sentence ‘The purpose of this study was to explore the association...’ could be revised for clarity. Suggest: ‘This study investigated the association between the risk of low energy availability, eating disorders and anxiety in female college athletes.’
Line 13 (age of female athletes): The age of female athletes is expressed with a standard deviation (‘19.9±0.1’). It might be more appropriate to round to a figure more understandable to the average reader, e.g. ‘19.9±1 years’.
Lines 26-42 (Introduction): Good introduction, but I recommend further development of the part about the clinical implications for early intervention. Include a paragraph emphasising how these findings can influence the prevention and clinical management of athletes to improve their physical and mental health.
Lines 61-68 (Participants): Although ethics committee approval is indicated, it might be useful to mention any exclusion criteria adopted to select participants. This would improve the transparency of the selection process.
Lines 91-92 (DESA-6 validation): The sentence ‘The DESA-6 is shown to be reliable and valid...’ should be revised for a more formal wording: ‘The DESA-6 has been validated as a reliable and valid instrument for measuring the risk of eating disorders in athletes.’
Lines 132-144 (Data analysis): It would be useful to specify whether robustness tests were performed to verify the stability of the results in case of small variations in the collected data. This would add credibility to the robustness of the results.
Lines 173-176 (Table 2, differences in psychometric scores): In the description of the results, a more detailed analysis of the clinical implications of the scores would be appreciated. I suggest elaborating briefly on the significance of these differences for the affected female athletes.
Lines 196-201 (Table 3): Consider including a visual interpretation of the data, such as a bar graph or scatter plot, to facilitate understanding of the correlations that emerged between the different parameters.
Limitations and Parts to be Eliminated:
Lines 289-311 (Discussion of limitations): The authors did not elaborate on the potential bias due to the self-selection of participants or the fact that the study was only conducted in women. I would suggest more emphasis on how the specific nature of the sample (only female college athletes) may influence the generalisability of the results. Furthermore, eliminating any repetitions in the limitations already discussed in the previous paragraphs might make the reading more fluent.
Lines 305-308: Delete or rephrase the sentence ‘it is currently unknown whether low energy availability causes anxiety...’ to avoid redundancy, as such uncertainty was already discussed above.
Limitations of the study (not discussed by the authors):
Causality: As highlighted, the study is cross-sectional in nature, therefore it is not possible to establish the causal direction between low energy availability and anxiety.
Sample size: The sample of only 115 participants limits the possibility of generalising the results to all female university athletes or, more generally, to all female athletes.
Lack of follow-up: It is unclear whether female athletes were monitored over time. A longitudinal study would have offered more information on the development of the psychological and physical condition of the participants.
Reviewer 2 Report
Comments and Suggestions for Authors
Thank you for your work. A research done with great effort. There are some shortcomings. By eliminating these, research will be in a better position.
Enjoy your work. I wish you success.
Abstract
Line 20-22: ‘’ In female collegiate athletes, both disordered eating and risk of low energy availability were associated with increased anxiety.’’ What was the direction of the relationship? Positive or negative? should be emphasized.
Introduction
The importance of the research should be addressed more clearly. Also, what was the main hypothesis of this study? The hypothesis should be included in the introduction before writing the purpose statement.
Materials and Methods
The model of the research was not stated. It should be written.
It is stated in the summary that the research was conducted on 115 people. There is no detail regarding this information in the "Participants" section of the method section.
Why what was done with 115 people? How was this number determined? Is this sample size sufficient? For this, a power analysis must be done.
Line 36: ‘’Independent Sample Mann Whitney U was used to…….’’ Is the full name of the test written like this? Please review.
Results
In Table 2, the less than sign is used for some p values, but not for others. Follow a standard spelling rule.
While interpreting Table 3, also express the direction of the relationship.
Conclusions
Before presenting the results and recommendations of the research, the limitations of the research should be expressed.
Reviewer 3 Report
Comments and Suggestions for Authors
Dear Corresponding Author, thank you for submiting your article to Sports journal and congratulations on your work.
Brief summary
This cross-sectional study examines the association between the risk of low energy availibility, eating disorders and anxiety in 115 female university atheletes. Using a series of validated questionnaires (DESA-6, EDE-QS, LEAF-Q and GAD-7), the study reveals significant correlations between anxiety, disordered eating behaviors and risk of low energy availability. The work contributes in an interesting and at times innovative way to understanding the interrelations between mental and physical health in university athletes.
General comments
The manuscript adresses a highly relevant topic in the field of sports medicine and sports psychology. The methodological aproach is solid and well-structured, with an appropriate selection of assessment tools. The statistical analysis is appropriate and well conducted, providing significant and interpretable results.
A particular strength of the study is the exploration of the independent relationship between anxiety and risk of low energy availibility, beyond the already known association with eating disorders. This offers new perspectives for understanding and managing athletes' health.
However, there are some areas that could benefit from further clarification or elaboration:
- The discussion could be expanded to explore more deeply the practical implications of the results for coaches, sports physicians and sports psychologists.
- It would be interesting to see a more in-depth reflection on possible differences between sports, even if the sample might be too limited for robust statistical analysis.
- The section on study limitations could be expanded, particularly regarding the cross-sectional nature of the study and limitations in inferring causality.
Specific comments
- Line 62-71: The description of the recruitment process could benefit from more details. For example, how were coaches and athletic directors initially contacted? Were there specific criteria for selecting the universities involved? What was the total number of reference athletes out of the 115 selected? Was there a drop-out? Are really "athletes"? What criteria or threshold was established to determine an individual's status as an 'athlete'? How many did not meet the inclusion criteria? In my opinion, this data is relevant.
- Line 172: Table 2 provides valuable information, but it might be useful to add a column with p-values for the differences between groups, to facilitate immediate understanding of statistical significance.
- Line 216-237: This section of the discussion is particularly interesting, but could benefit from further elaboration. You might consider expanding the discussion on possible physiological mechanisms linking low energy availability and anxiety.
- Line 274-288: The practical implications discussed here are important, but could be further developed. For example, could you suggest specific screening or intervention strategies based on your findings?
- Line 289: you have addressed the limitations very briefly, I believe there are many other critical areas that deserve to be pointed out, such as the absence of actual measurements on body parameters (weight, type of sport in the absence of specific data on type of previous experience etc), all relying on "self-reportered" when, especially in the female field, some data tend to be underestimated as well present in the literature, for example https://doi.org/10.1111/j.1467-789x.2007.00347.x
- Line 305-314: The conclusions are well written, but you might consider adding a sentence or two on the long-term implications for athletes' health and performance, based on your findings.
In general, the manuscript is well written and structured, with a clear logical flow from introduction to conclusions. The tables are informative and well presented.
However, I believe that excessively positive conclusions are reached, underestimating very strong criticisms of the study, which the authors could probably correct. With some major revisions as suggested, this article will be a significant contribution to the literature on university athletes' health and for Sports journal.
I remain curious and interested in receiving an improved version.
Round 2
Reviewer 1 Report
Comments and Suggestions for Authors
I would like to inform you that the article submitted, following the modifications made, is now at a stage where it can be further revised. The requested changes have been satisfactorily integrated, improving both the clarity and coherence of the text. However, as is often the case in these processes, there is still room for further refinement, especially regarding certain stylistic and content aspects that could benefit from a final revision before the definitive publication.
I therefore invite you to consider the opportunity for a thorough rereading, with the aim of further perfecting the work and ensuring that the message conveyed fully aligns with your intentions and the journal's expectations. In this regard, it is always useful to carefully check the flow of the argument, the accuracy of the cited sources, and the internal consistency of the discourse, in order to present readers with a high-quality text.
I remain available for any clarifications or further interventions and thank you for your collaboration up to this point. I wish you all the best in the final phase of this revision.
Author Response
Thank you for the comments.
Reviewer 2 Report
Comments and Suggestions for Authors
Thank you for your responses and edits. Enjoy your work.
Author Response
Thank you for your time reviewing our work.
Reviewer 3 Report
Comments and Suggestions for Authors
I have thoroughly reviewed the revised version of this manuscript. I believe it is a good article with modest interest for readers, but I don't see any limitations to its publication in its current form.
Author Response

(The authors gave the same response as above.)
